# Socio-economic and environmental factors affecting breastfeeding and complementary feeding practices among Batwa and Bakiga communities in south-western Uganda

Giulia Scarpa[1,2]*, Lea Berrang-Ford[1,3], Sabastian Twesigomwe[3], Paul Kakwangire[3], Maria Galazoula[4], Carol Zavaleta-Cortijo[5], Kaitlin Patterson[6], Didacus B. Namanya[3,7], Shuaib Lwasa[3,8,9], Ester Nowembabazi[3], Charity Kesande[3], IHACC Research Team[3], Janet E. Cade[2]

1 School of Environment, University of Leeds, Leeds, United Kingdom, 2 School of Food Science and Nutrition, University of Leeds, Leeds, United Kingdom, 3 Indigenous Health Adaptation to Climate Change Research Team, Kanungu District, Buhoma, Uganda, 4 Leeds Institute for Data Analytics, University of Leeds, Leeds, United Kingdom, 5 Facultad de Salud Publica y Administracion, Universidad Peruana Cayetano Heredia, Lima, Peru, 6 Department of Population Medicine, University of Guelph, Guelph, Canada, 7 Ministry of Health, Kampala, Uganda, 8 Department of Geography, Makerere University, Kampala, Uganda, 9 The Global Center on Adaptation, Rotterdam, Netherlands

* eegs@leeds.ac.uk

**Data Availability Statement:** All data available to share publicly are contained within the article, and

## Abstract

Improving breastfeeding and complementary feeding practices is needed to support good health, enhance child growth, and reduce child mortality. Limited evidence is available on child feeding among Indigenous communities and in the context of environmental changes. We investigate past and present breastfeeding and complementary feeding practices within Indigenous Batwa and neighbouring Bakiga populations in south-western Uganda. Specifically, we describe the demographic and socio-economic characteristics of breastfeeding mothers and their children, and individual experiences of breastfeeding and complementary feeding practices. We investigate the factors that have an impact on breastfeeding and complementary feeding at community and societal levels, and we analysed how environments, including weather variability, affect breastfeeding and complementary feeding practices. We applied a mixed-method design to the study, and we used a community-based research approach. We conducted 94 individual interviews (n = 47 Batwa mothers/caregivers & n = 47 Bakiga mothers/caregivers) and 12 focus group discussions (n = 6 among Batwa & n = 6 among Bakiga communities) from July to October 2019. Ninety-nine per cent of mothers reported that their youngest child was currently breastfed. All mothers noted that the child experienced at least one episode of illness that had an impact on breastfeeding. From the focus groups, we identified four key factors affecting breastfeeding and nutrition practices: marginalisation and poverty; environmental change; lack of information; and poor support. Our findings contribute to the field of global public health and nutrition among Indigenous communities, with a focus on women and children. We present recommendations to improve child feeding practices among the Batwa and Bakiga in south-western Uganda. Specifically, we highlight the need to engage with local and national authorities to improve

no further information about qualitative data can be shared due to ethical/privacy reason as we worked with vulnerable communities. Queries about the underlying data may be sent to Jennifer Blaikie (Senior Research Ethics Administrator at the University of Leeds) at j.m.blaikie@leeds.ac.uk, researchethics@leeds.ac.uk.

**Funding:** G.S. was financed by a scholarship from the Canadian Institute of Health Research. This work is part of a larger project, the Indigenous Health and Adaptation to Climate (IHACC). Financial support for that project is provided by the International Development Research Centre, Tri-Council Initiative on Adaptation to Climate Change, IHACC, IDRC File nos. 106372–003, 004, 005. The funding sources had no role in the study design, data collection, analysis, interpretation of data. C. Z.-C. was supported by the National Institute for Health Research (NIHR) (using the UK's Official Development Assistance (ODA) Funding) and Wellcome [218743/Z/19/Z] under the NIHR Wellcome Partnership for Global Health Research. The views expressed are those of the authors and not necessarily those of Wellcome, the NIHR or the Department of Health and Social Care.

**Competing interests:** I have read the journal's policy and the authors of this manuscript have the following competing interests: J.E.C. is Director of Dietary assessment Ltd. No other competing interests to declare.

breastfeeding and complementary feeding practices, and work on food security, distribution of lands, and the food environment. Also, we recommend addressing the drivers and consequences of alcoholism, and strengthening family planning programs.

## Introduction

In 2019, 21% of children under 5 years globally were stunted and 2% wasted [1]. Undernutrition in infants exacerbates the risk of mortality, morbidity, and chronic diseases, and causes delays in neuro-psychomotor development [2,3]. Poor children living in vulnerable settings are 20 times more at risk of undernutrition than others [1], and especially among Indigenous communities [2,4]. This is caused by economic disparities, socio-cultural discrimination, and colonial legacies that translate into health inequalities [5].

Research demonstrates that improvements in infant and young child feeding (IYCF), and traditional feeding practices are critical to supporting good health, enhancing child growth, and reducing child mortality [6]. Breastmilk is the only nutritional source recommended by the WHO for newborns and infants up to 6 months [7]. According to WHO statistics, however, only 1 out of every 3 children worldwide is exclusively breastfeed for their first six months of life, and only 2 of 5 are immediately breastfed in the first hour after birth [8]. In addition, the WHO and UNICEF recommend starting complementary feeding along with breastfeeding after the first 6 months of life to avoid stunting in childhood [9,10]. Breastmilk alone is insufficient to ensure adequate child growth after 6 months of age [2]. Low quality and quantity of foods, and late introduction of solid foods are found to be causes of undernutrition [11].

Maternal diets influence infant diets. Tiedje et al. [12] proposed an ecological approach to understand the influences of maternal nutrition on breastfeeding and infant feeding practices by analysing contextual factors such as family, community and healthcare system. Later, others extended the focus of this model to also include societal contexts and changing environments [13]. Indeed, child feeding has been observed in Indigenous populations [14] to be driven by social and environmental contexts, which were identified as critical in understanding breastfeeding and infant feeding practices. For example, Sellen documented changes in individual choices and food behaviour among Indigenous families due to changes in socio-environmental conditions, such as culture, work activities, natural environment, and traditional food supplies [15]. In some cases, mothers substituted traditional infant foods with packaged food during the complementary feeding period [16].

While there are well-established indicators to measure changes in child nutrition linked to health (breastfeeding duration, starting age of complementary feeding, minimal dietary diversity, minimum meal frequency, minimum acceptable diet), few studies have assessed IYCF among Indigenous groups or in the context of environmental change [14,17]. The aim of this study was to investigate past and present breastfeeding and complementary feeding practices within the Indigenous Batwa and Bakiga populations. The main objectives were: 1. To describe the demographic and socio-economic attributes of breastfeeding mothers and their children, and individual experiences of breastfeeding and complementary feeding practices; 2. To investigate the factors that have an impact on breastfeeding and complementary feeding at community and societal level; 3. To analyse if and how environments, including weather variability, affect breastfeeding and complementary feeding practices.

## Material and methods

Ethics approvals were obtained from the University of Leeds Research Ethics Board (AREA 18–156), the Ugandan National Council for Science and Technology (SS5164), and the Makerere University Research Ethics Committee (MAKS REC 07.19.313/PR1). For the minors included in the study, we obtained consent from their parents or guardians.

### Study population

The Batwa and Bakiga communities live in the District of Kanungu in south-western Uganda, located on the border with the Democratic Republic of Congo. The Bakiga are historically an agrarian society, who depend on agriculture and livestock, and represent the majority of the population [18]. Both populations suffer from high burden of illness, and especially the Batwa have high incidence of malaria, malnutrition and gastrointestinal diseases [19–21], while the Bakiga have higher levels of HIV [22]. From previous studies, there is evidence that malnutrition is very high among Batwa communities, especially among children under five: 8% of male children classified as wasted [18], for example, compared to 4% nationally [23]. The proportion of male undernourished cases among Batwa is higher than among females, with boys at greater odds of being severely malnourished. In every Batwa age-sex grouping, 15% or more individuals are malnourished [20]. Also, according to *Patterson et al.* [24], more than 90% of Batwa households are rated as "very highly food insecure". In 2017, Batwa mothers reported being malnourished and having malnourished children due to a scarcity of food [25].

### Conceptual framework

We adapted the conceptual framework of Hector et al. [13] (Fig 1). Our framework is composed of four levels: individual, group, societal and environmental. The individual level analyses the characteristics of the mother, the child, and the dyad (mother-child pairing), including knowledge on breastfeeding, mother-child interactions, and health status. The group level describes factors that influence breastfeeding practices in their proximal social setting, for example accessibility to health facilities, and mothers' work and community environments. The societal level considers contextual elements that have an impact on breastfeeding choices such as the role of women and men in society, and cultural norms [13]. Finally, we included an environmental level to the framework to further analyse the impact of environmental contexts and events (weather, land suitability for food production, and extreme events such as flooding or drought) on breastfeeding and complementary feeding practices.

Marginalisation of Indigenous Batwa communities is present at all levels of society. The Batwa represent 1% of the Ugandan population, and were displaced in 1991 by the Ugandan Government from their ancestral forest lands to create the Bwindi Impenetrable National Park [26]. Traditionally hunters-gathers, since 1991 the Batwa have begun transitioning to cultivating crops [24]. They work mostly as farmers, hired by the Bakiga; some are craft makers or brick makers or work in tourism [18]. However, the absence of a traditional culture of farming and low socio-economic status exacerbate food insecurity [27] which have an impact on individual nutrition, and can affect negatively child health [28]. Although many Batwa and Bakiga families are poor, Batwa per capita income is substantially lower than the national average (0.36 US dollars/day compared to 0.99 US dollars/day) [24]. Inequities are also found in education. Less than 12% of Batwa living in Kanungu District are able to write and read, and to access education [24]; the school drop-out rate is especially high following marriage at a young age [29].

Persistent poverty underpins poorer access to healthcare services among Batwa, despite similar health facilities for both populations [20]. For example, compared to the rest of Uganda

**BREASTFEEDING AND COMPLEMENTARY FEEDING PRACTICES**

**Fig 1. Conceptual framework used to analyse breastfeeding and complementary feeding among the Batwa and Bakiga communities.** The group level refers to the level of household and community; the societal level refers to the level of region, ethnicities and country. There is interaction across all levels; for example, the environmental level interacts with the societal and group levels.

where 57% of mothers gave birth with health professionals, only 40% of Batwa births occurred in health facilities in 2017 [22]. Also, in cases of child malnutrition, Batwa mothers tend to stay at home rather than go to the hospital for treatment [20]. One of the main reasons for scarce hospital attendance is the cost of health insurance premiums [30], which are not affordable for many Batwa families, as well as persistent social and ethnic discrimination [18]. Also, Batwa more frequently report not having soap or access to health facilities compared to non-Indigenous neighbours, with negative consequences on maternal and child health [31].

Poverty and discrimination are linked to poor mental health, alcoholism and domestic violence.. This link is documented in previous studies, although the research on mental health among the Batwa and Bakiga communities remains limited [24]. Previous research indicates that internal displacement, as occurred among the Batwa population, can have a negative effect on health, wellbeing and socioeconomic status [32]. Batwa and Bakiga report high levels of alcoholism, particularly among men and during periods of food insecurity. High consumption of alcohol is an emerging problem among these communities, and the local hospital has addressed this by offering alcohol rehabilitation services [33]. Alcoholism has been linked to domestic violence [24,34]. Reported implications of alcoholism among Batwa include compromised food security for adults and children, mental health concerns, and poorer health outcomes [24].

Poverty and marginalisation are projected to be exacerbated by the impacts of climate change with consequences on nutrition and health, especially for young children [35,36]. The District of Kanungu is affected by extreme climatic events, such as the floods that occurred in 2019. Climate change projections anticipate an increase in annual mean temperature and change in seasonal variation [37,38]. The dry season is usually between December and February, and between June and August, however rainfalls are now longer (from September to December, and then from March to June) and less predictable, with fewer sunny days [39].

These changes are impacting key local crops (groundnuts, beans and cassava) that are an important nutritional source for the Batwa and Bakiga communities [39].

## Study design

We undertook a mixed-method study to explore breastfeeding and complementary feeding practices among mothers with children under two years in Kanungu district.

The research was guided by a community-based participatory research approach which engage researchers and community participants as equal partners during the research process; the objective is to educate or promote social change [40,41]. This approach has been used among marginalised groups, especially Indigenous communities, as it helps to reinforce the respect and collaboration between stakeholders and researchers [42].

Batwa and non-Batwa participants assessed the barriers to breastfeeding and complementary feeding practices, evaluated and shared their level of knowledge on maternal and infant nutrition, and critically explored solutions to improve mother and child health.

## Settlement and individual sampling

Twelve communities, six Batwa and six geographically matched adjacent Bakiga settlements, were included in the study (Fig 2).

The selection of communities sought a sample representing variation in terms of geographic location and market access. Mothers with children aged under two years were sampled from 12 settlements (6 Batwa and 6 Bakiga): 1) two settlements located very close to the market and shops (Bikuuto/Bikuto cell and Kihembe/Kengoma cell), 2) two settlements close to the forest and located very far from the market and shops (Kitariro/Kitariro cell and Mpungu/Kikome cell area), and two settlements situated mid-way (Kebiremu/Kebiremu cell and Byumba/Byumba cell). The selection of samples in the different areas allowed for exploration of geographically linked variation in practices and availability of food for infants' complementary feeding between Batwa and Bakiga communities and between settlements with different proximity to market centres.

Twelve focus groups were conducted, one in each of the communities (n = 6 Batwa and n = 6 Bakiga), with 7–11 mothers per group. First, the research assistant met the local leader of each community, sought the permission to conduct the study and asked for a list with the names of mothers with children under 2 years living in Kanungu District. We invited all eligible Batwa mothers to participate, as the number of mothers was limited (47 individuals). In the case of the Bakiga, we conducted a random sample of eligible mothers, as the number of women with children under 2 years was greater. A number was consecutively assigned to each eligible individual starting at one. A table of random numbers was used to sample the 47 individuals who were invited to participate [43].

## Data collection

Three Ugandan researchers collected the data from July to October 2019. The research team was composed of a Ugandan male researcher living in Kanungu district, and a Mutwa (Batwa singular) and a Mukiga (Bakiga singular) female researcher. Focus group discussions (FGDs) and the individual interviews were conducted and audio-recorded in the local language, Rukiga, and then the information was translated into English by the local researcher (ST). Before sampling individuals, the research team contacted the chairperson of each community to introduce the study and seek his/her approval. Written consent was obtained before all interviews.

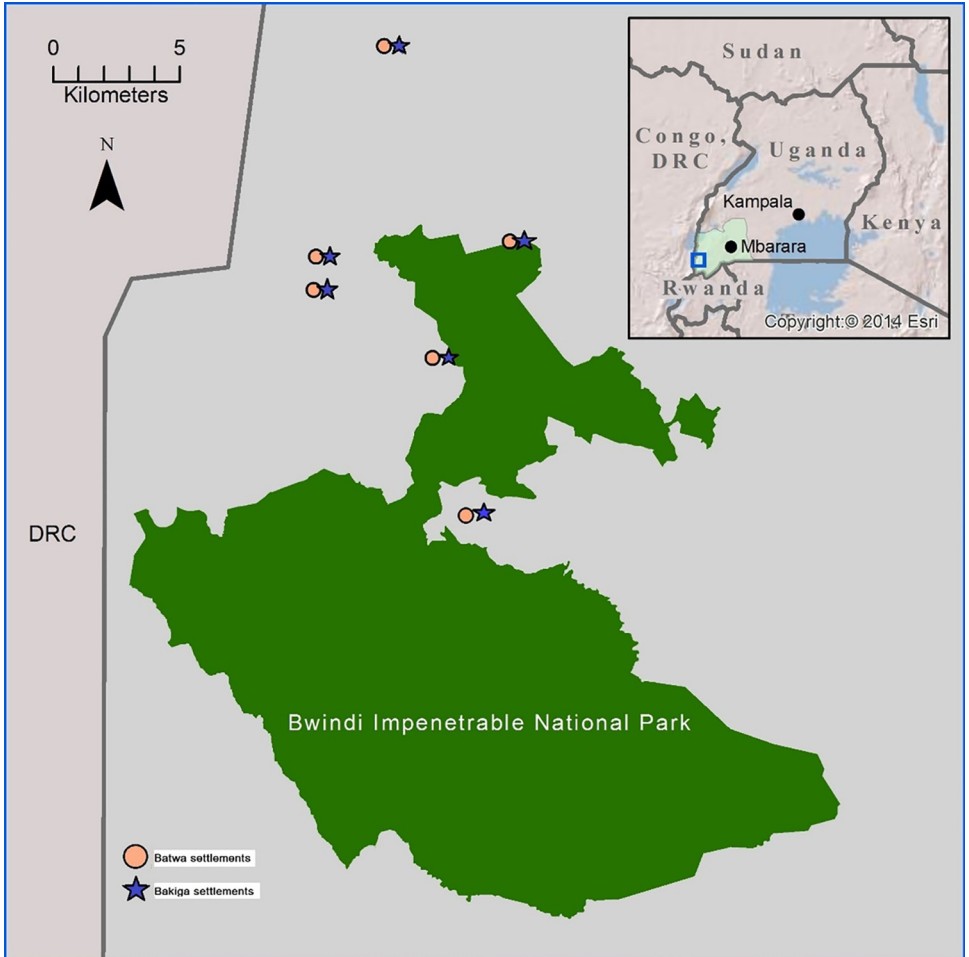

**Fig 2. In the map we represented the ten Batwa settlements that participated in the research; there is a correspondent Bakiga settlement for each Batwa settlement.** In our study, we involved 6 Batwa and 6 Bakiga settlements: Bikuuto (Batwa) & Bikuuto cell (Bakiga), Kihembe (Batwa) & Kengoma cell (Bakiga), Kitariro (Batwa) & Kitariro cell (Bakiga), Mpungu (Batwa) & Kikome cell (Bakiga), Kebiremu (Batwa) &Kebiremu cell (Bakiga) and Byumba (Batwa) & Byumba cell (Bakiga). Map adapted from Patterson, 2017 [24]. Coordinates of settlements described in S1 Data.

We created an interview guide with open questions for the FGD (S1 and S1A Text). The questions followed the themes from the Optimal IYCF guidelines, specifically the complementary feeding section [9,10]; IYCF programmes aim to prioritize and improve breastfeeding and complementary feeding practices to reduce malnutrition worldwide [9]. FGDs helped to explore the complexity of lived experiences that were not capturable with standard questionnaires, and encouraged women to share knowledge and perceptions on IYCF. Each group discussion lasted on average 62 minutes.

The individual interview questionnaire included primarily closed questions, with some requiring brief explanation (S2 and S2A Text). Some were adapted from the Standardized Monitoring and Assessment of Relief and Transition Methodology [44] to evaluate some factors linked to the nutritional status such as hygiene, food and water accessibility level. Interviews lasted on average 12 minutes.

## Data analysis

To respect the multi-perspectivity and the multivocality of the data [45–47], we coded, analysed and organised the qualitative data using NVivo 12® software. Contextualized thematic analysis (latent and manifest) was used to analyse the data [48]. The analysis process involved data familiarization, generating initial codes, defining, reviewing, and naming themes. In this manuscript, we reported the findings of the themes that were discussed by the participants during the FGDs as key factors of successful or unsuccessful breastfeeding and complementary feeding practices. Additionally, we descriptively analysed demographic data on Batwa and Bakiga women and children using Stata® version 15.

## Inclusivity in global research

Additional information regarding the ethical, cultural, and scientific considerations specific to inclusivity in global research is included in the Supporting Information (S3 Text).

# Results

## Characteristics of mothers/child caregivers and children

Ninety-four women (47 Batwa and 47 Bakiga) took part in the individual interviews and focus group discussions (FGDs). A description of the 94 mothers/child caregivers and 95 child participating in the individual interviews included data on attributes and interaction between mothers and children and can be found in Table 1. The response rate in both FGDs and individual interviews was 100%. In the case of four Batwa children, the participating primary carer was a grandmother rather than mother (e.g. the mother had died or left the settlement). The number of Bakiga children included in the study was 48 as there was one set of twins.

**Table 1. Description of the women and children participating in the study.**

| Sample: 94 women with 95 infants | Batwa women (N = 47) | Bakiga women (N = 47) |
|---|---|---|
| | *Counts (percentage)[1]* | *Counts (percentage)* |
| *Age of female primary caregivers* | | |
| 15–19 | 5 (11) | 4 (9) |
| 20–24 | 10 (21) | 13 (28) |
| 25–29 | 7 (15) | 17 (36) |
| 30–34 | 14 (30) | 5 (11) |
| > = 35 | 11 (23) | 8 (17) |
| *Community of residence* | | |
| Kebiremu/Kebiremu cell | 8 (17) | 7 (15) |
| Kitarir/Kitariro cell | 7 (15) | 7 (15) |
| Mpungu/ Kikome cell | 6 (13) | 7 (15) |
| Bikuuto/Bikuuto cell | 10 (21) | 7 (15) |
| Byumba/Byumba cell | 7 (15) | 8 (17) |
| Kihembe/Kengoma cell | 9 (19) | 11 (23) |
| *Number of children under 5 years in the household* | | |
| One child | 9 (19) | 11 (23) |
| Two children | 8 (17) | 9 (19) |
| Three children | 8 (17) | 9 (19) |
| Four Children | 8 (17) | 8 (17) |

*(Continued)*

**Table 1.** (Continued)

| Sample: 94 women with 95 infants | Batwa women (N = 47) | Bakiga women (N = 47) |
|---|---|---|
| | *Counts (percentage)[1]* | *Counts (percentage)* |
| More than four children | 14 (30) | 10 (21) |
| *Mothers who lost at least 1 child during pregnancy or after birth* | 20 (43) | 8 (17) |
| *Birth interval (referred to last child)* | | |
| < 24 months | 2 (4) | 3 (6) |
| > = 24 months | 45 (96) | 44 (94) |
| *No illnesses/complications during child birth[2]* | 41 (87) | 40 (85) |
| *Received food from Government/NGOs* | 43 (91) | 5 (11) |
| *Own land (small size)* | 44 (94) | 47(100) |
| *Own animals[3]* | 19 (40) | 39 (83) |
| *Pit latrine access* | 47 (100) | 47 (100) |
| *Access to soap* | | |
| Yes | 2 (4) | 30 (64) |
| Sometimes | 32 (68) | 16 (34) |
| No | 13 (28) | 1 (2) |
| *Protected water facilities* | 39 (83) | 47 (100) |
| *Age at first pregnancy* | *Mean/(SE)* 18.9 (0.4) | *Mean/(SE)* 19.5 (0.4) |
| *Child age* | **Batwa children (N = 47)** | **Bakiga children (N = 48)** |
| 0–6 months | 20 (43) | 9 (19) |
| 7–12 months | 13 (28) | 12 (25) |
| 13–18 months | 8 (17) | 9 (19) |
| 19–23 months | 6 (13) | 18 (38) |
| *No illnesses/complications at birth[4]* | 45 (96) | 44 (92) |
| *Currently breastfed* | 46 (98) | 48 (100) |
| *Child looked after by the mother/caregivers only* | 40 (85) | 38 (79) |
| *Place of birth* | | |
| Home | 19 (40) | 7 (15) |
| Health Centre | 4 (9) | 17 (35) |
| Hospital | 24 (51) | 24 (50) |
| *Reported child illnesses/symptoms in the first six months of life by caregivers* | | |
| Cough | 31 (66) | 32 (67) |
| Diarrhoea | 38 (81) | 35 (73) |
| Flu | 23 (49) | 20 (42) |
| Fever | 7 (15) | 8 (17) |
| Malaria | 40 (85) | 20 (42) |
| Pneumonia | 15 (32) | 11 (23) |
| Vomiting | 8 (17) | 4 (8) |
| Skin diseases | 4 (9) | 8 (17) |
| Other | 8 (17) | 5 (10) |

[1]The percentages were calculated over 47 participants for both Batwa women and children and Bakiga women, and over 48 for Bakiga children.

[2]Illness/complications refer here to post-partum haemorrhage, retained placenta and death. Death occurred in 1 case among the Batwa participants.

[3]Animals include chickens, rabbits, goats, pigs and cows.

[4] Illnesses/complications refer here to respiratory problems needed resuscitation management at birth.

From the data collected through the individual interviews, there were some differences in demographic and health characteristics between Batwa and Bakiga, primarily related to wealth/assets, sanitation/hygiene, and place of birth. The majority of Batwa did not have regular access to soap (32 of 47 participants, 68%); although most of the participants reported washing their hands with water only before and after breastfeeding, after visiting the toilet and eating, none reported hand washing before cooking or preparing food for the children and family. Eighty-three percent (n = 39) of Bakiga mothers reported owning at least one animal compared to 40% (n = 19) of Batwa mothers. During the interviews, most Batwa mothers (91%, n = 43) reported receiving food aid from NGOs or government, while only a few Bakiga mothers (11%, n = 5) living close to the forest reported receiving compensation in the form of food (when elephants destroyed their gardens).

Children of Batwa mothers in this study were generally younger (n = 20 or 43% of children aged 0–6 months) compared to children of Bakiga mothers (n = 18 or 38%). Births among Batwa mothers occurred more frequently at home (40%, n = 19) compared to Bakiga mothers (15%, n = 7). Reporting of child loss during pregnancy and after birth was higher among Batwa (43%, n = 20) compared to Bakiga (17%, n = 8) mothers.

Mothers reported that nearly 99% of children were currently breastfed, and were able to breastfeed at their place of work or livelihood activity, typically their family farm. Mothers of all children older than 7 days reported that their child had experienced one or more episodes of illness in the first 6 months of life that had had a negative impact on breastfeeding.

## Main findings from the focus group discussions

We identified the key factors constraining child feeding through the focus group discussions which are summarised in Table 2. The elements are interlinked and positioned across the four levels of the framework (Fig 1): marginalisation and poverty at group/societal level, crossing the environmental and individual level; environmental changes at environmental level, crossing the societal level; lack of breastfeeding and complementary feeding-related information at individual level, crossing the group/societal level; and poor support at group level, crossing the societal level.

**Marginalisation and poverty.** The Batwa and Bakiga participants identified poverty as the dominant driver limiting the success of breastfeeding and complementary feeding practices. They repeated during the discussion that "*we don't have enough money, we are poor, thus we cannot feed our children*" (Batwa FGD), and drew a link between lack of money and resources, and poor nutrition outcomes:

**Table 2. Summary of the key factors affecting breastfeeding and complementary feeding according to the participants in the FGD.** For each key factor we identified the correspondant level(s) explained in the framework (Fig 1).

| Key factors | Level(s) | Description |
|---|---|---|
| Marginalisation and Poverty | Group & Societal | Due to marginalisation and high poverty levels, Batwa and Bakiga participants cannot afford to buy enough and high-quality foods. This had a negative impact on child nutrition and growth. |
| Environmental changes | Environmental | Extreme climatic events, especially droughts and floods, negatively impacted food security, especially accessibility and availability of foods with consequences on child and maternal nutrition. |
| Lack of breastfeeding and complementary feeding information | Individual | Lack of information received from health workers on complementary feeding practices and cultural beliefs were found to be limiting factors to achieve good nutrition in children according to the Batwa and Bakiga mothers. |
| Poor support | Group | Support is fundamental during breastfeeding and complementary feeding practices. However, this was limited for the Batwa and Bakiga women, who were used to live in contexts of domestic violence and alcoholism. |

*"If you are poor, and you don't eat enough, you become sick. If mother and baby cannot eat properly, they both become sick and malnourished [. . .] The mother cannot produce enough (milk), she is very weak, has dizziness, is dry, so she cannot produce milk and the baby cries"* (Batwa FGD).

Batwa mothers added that the Bakiga are also impoverished: *"not only the Batwa do not have food, but also the Bakiga, so the Batwa cannot really work for them for food"* (Batwa FGD). Batwa and Bakiga participants talked about their eviction from the forest, their traditional land, where previously they looked for food, now forbidden by the Government; they linked their historic displacement—and persistent restriction of forest access—to their current poverty and food insecurity:

*"Most of the women do not have enough milk when the baby is older than 1 year, because they do not have millet porridge and other helpful foods such as cassava leaves. You can have enough breastmilk if you eat porridge, meat and fish [. . .]. But we don't have access to this food now because we cannot go to the forest, and we don't have money to buy it"* (Batwa FGD).

Also, *"In the past children started on wild meat, yams, mushrooms, greens from the forest as we were allowed to enter. Now we don't have access, so we use what we can afford"* (Bakiga FGD).

During the discussion, mothers also gave advice on specific foods that pregnant and lactating women should eat to increase milk production: millet porridge, greens, beans, Amaranthus leaves, and cassava leaves (Batwa FGD), and some Bakiga participants added also the consumption of meat and fish.

Women explained that children have a greater risk of malnutrition due to maternal poor diet, and infants are more likely to get sick with vomiting and diarrhoea due to 'low-quality' food, poor in proteins. They agreed that the quality of nutrition depends on money availability: *"only if we have money, we can buy eggs, which are very nutritious for the baby"* (Bakiga FGD). The most common foods for complementary feeding listed by mothers, and given from 4–5 months of age are matoke (green bananas), sweet or Irish potatoes, cassava, groundnuts, soup from meat (if available), and beans.

A Mutwa woman highlighted how their situation has changed compared to the past in terms of food availability: infants now are fed with what is available at home, and they don't always have 'sauce' (beans, groundnuts or greens) to accompany cereals. According to the participants, poor quality food causes sickness in both mothers and children:

*"If the woman is sick, this means that even the child won't eat as the mother cannot go to work, and without working there is no food for that day"*.

Participants added that appropriate complementary feeding practices get more complicated to follow in case of twins or multiple children under 5 years of age as it is difficult to provide sufficient foods to satisfy their nutritional needs. Also, they argued that is harder to produce enough milk for two or more children at the same time, therefore weaning usually starts before 4 months (Bakiga FGD).

Participants suggested that improving food availability and land accessibility through collaboration with the Government and district would have a positive effect on dietary diversity,

*"to have a balanced diet"*, and increase the quantity of food consumed by household (Batwa and Bakiga FGD).

The participants explained that food insecurity is also a consequence of limited availability of lands, and overpopulation; therefore, food production is decreasing with negative effects on child nutrition:

> *"The land is not fertile, there is no sorghum or honey available that we gave before to babies [. . .] We need more land, our pieces of land are small and the population is growing."* (Batwa FGD).

> *"We don't have enough land to grow nutritious food for the children, and the little we have, we have to share among families and none get satisfied from the food"* (Bakiga FGD).

For this reason, the women addressed the need of more lands to distribute among the community, and they sought to assess and redress barriers related to insecure land tenure and poor land quality to increase food production and availability (Batwa and Bakiga FGD).

**Poor support.**   Batwa and Bakiga participants reported that they lack support from partners, the community, and health workers during the breastfeeding and complementary feeding period.

The women in the study identified the role of partners and older children as important in supporting successful breastfeeding and complementary feeding. They suggested that older children usually help mothers to collect food for the family and look after the younger siblings when the mother works or is out of the home (Batwa and Bakiga FGDs). According to participants, partners (generally a husband) provide support by purchasing food for the family, assisting during feeding the child when ready for complementary foods, but partners' support varies from household to household (Batwa and Bakiga FGDs). They also added that if the father does not buy food for the family, there may be a delay in the start of complementary feeding even in the case of poor breastmilk supply, leading to a deterioration in child nutrition (Bakiga FGD). For this reason, women requested more economic and social support from men, including help from the local district authorities to reduce alcohol consumption (Bakiga FGD).

In fact, participants indicated that domestic violence (verbal and/or physical) is exacerbated by high rates of alcoholism in the communities, in particular among men but also, in lower percentage, among women; they suggested to ask for the help of the district and Government to as this is also disruptive to child feeding:

> *"Alcohol is a big problem [. . .]. Some men spend money on alcohol and do not provide food for the family. Sometimes there are also women that drink, and this can also cause miscarriages if they fight when they are pregnant [. . .] Also, women cannot produce enough milk for the baby if they are stressed"* (Omutwa participant).

The participants articulated their perspectives on the causes of alcoholism and domestic violence, suggesting that alcoholism and domestic violence are driven by extreme poverty, inequalities and general and persistent food insecurity (Fig 3):

> *"[..] men drink also if they are very hungry, they have to fill their stomach to feel satisfied, but when they drink alcohol, they can beat their wife"* (Bakiga FGD).

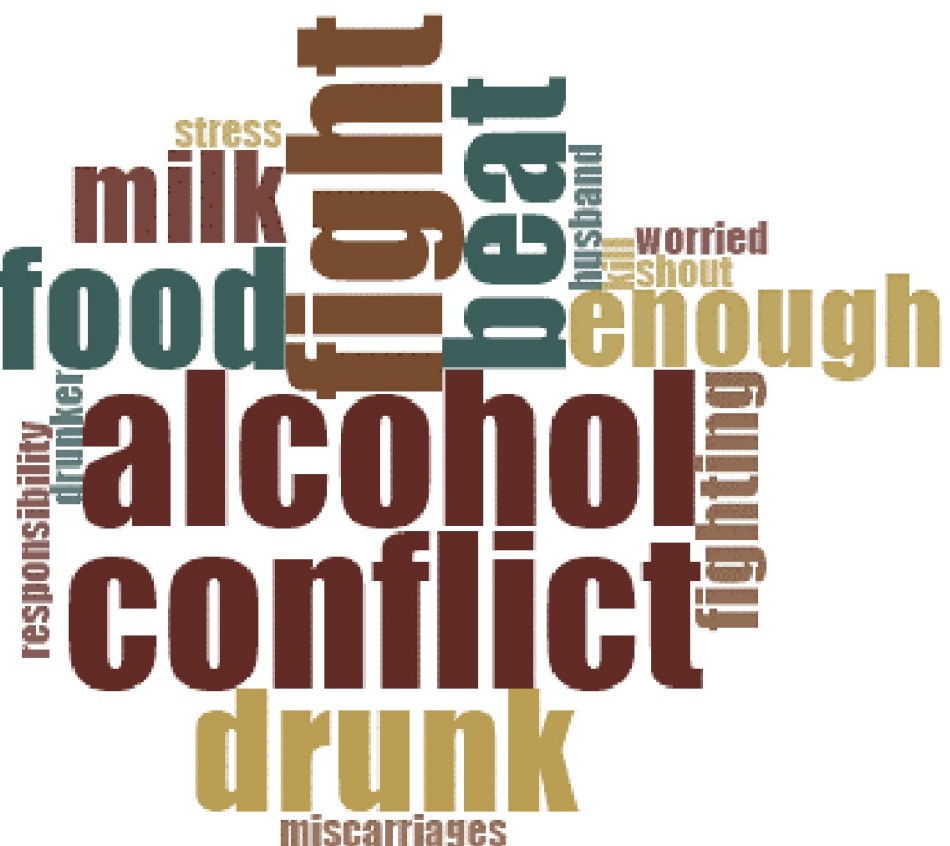

**Fig 3. Word frequency analysis with the most recurrent terms used by the participants to articulate the key drivers at the nexus of alcoholism, domestic violence, and child nutrition (breastfeeding and complementary feeding).** The bigger the word, the more often it was mentioned during the FGDs (Scale: 1–50).

Women mentioned that other support is usually offered by the community, and especially by relatives, friends, parents and neighbours. They usually provide food or help a mother to look after the baby and share information and knowledge on breastfeeding and weaning practices:

*"Usually parents or grandparents tell their daughter how to raise the child, which food to eat and to give to the child for complementary feeding, especially if this is the first child"* (Mutwa participant).

Participants also reported other conditions where mothers lack support for breastfeeding. Many mentioned that they are able to breastfeed while working, but they also argued that in some workplaces breastfeeding is not allowed, and they reported the case of sex workers:

*"Some educated mothers may not breastfeed the child because they cannot bring him/her at work; (in this case) the infant is left at home with the baby sitter who usually offers cows milk and porridge rather than breastmilk"* (Batwa FGD).

Additionally, Batwa women explained how breastfeeding was difficult among sex workers:

*"Usually (sex workers) are young and they are not ready to have a child. When they have a child, they do not have support, and they won't breastfeed the baby because this changes the body shape, and they need to keep beautiful for their work"* (Batwa FGD).

The women also noted that they lack strong support from health professionals to support mothers to breastfeed longer:

*"Batwa usually get pregnant fast, and for this reason they stop breastfeeding at 1 year. Some stop at 9 months or 6 months if they get pregnant, and we need more support from the hospital to end this"* (Batwa FGD).

Bakiga women highlighted the need to strengthen family planning services to help women not only to breastfeed longer, but also to decide when conceiving. According to the participants, young women and teenagers are in fact more likely to have their next child within 23 months due to lack of information and lower use of contraception. Specifically, they suggested to involve governmental institutions to address this need.

Additionally, Batwa and Bakiga participants argued that support is also needed to improve hospital and health centre accessibility as mothers face challenges in accessing medical services when they need assistance:

*"The problem is that the hospital is too far from here [..], there are mothers that lose their child because they need to walk too much or they do not reach [the hospital] in time. Also, once we reach it, we don't have food to eat in the days we stay there. There is no free service for us when we are hospitalised. And who [takes care of] the children that we left at home?"* (Batwa FGD).

**Lack of information on breastfeeding and complementary feeding.**    Both Batwa and Bakiga participants indicated that they had a good understanding of breastfeeding practices, and noted that it is healthy for their child and should be started as soon as possible after birth (Table 4). However, mothers felt that information on complementary feeding was insufficient. For this reason, they asked for more advice during antenatal visits and immunisations as in these circumstances the health professionals talk about breastfeeding in particular (Batwa FGD). Information on child feeding is provided to mothers by the hospital or NGOs (Table 3); the main topics covered in these sessions includes breastfeeding practices, complementary

**Table 3. Subjects covered by the hospital and NGOs related to breastfeeding and complementary feeding.**

| Courses mentioned by the Batwa and Bakiga participants | Course organiser |
|---|---|
| i) the principles of breastfeeding practices from positioning the baby to the breast to hygiene measures to apply | Hospital |
| ii) some information on the introduction of complementary feeding, and solid foods | Hospital |
| iii) the importance of delivering at the hospital with qualified health workers, taking supplementation during pregnancy and breastfeeding, and vaccinate the child at appropriate age | Hospital |
| iv) the causes of malnutrition, and how to avoid it | Hospital/NGOs |
| v) the importance of using family planning measures to control births and breastfeeding longer | Hospital/NGOs |
| vi) the need of testing the mothers for HIV before delivery (factor highlighted in the Bakiga FGD only) to know how long to breastfeed | Hospital |
| vii) the importance of microfinance courses to teach how to save or earn money by selling animals, and use it for transport to the hospital at the time of delivery or to cover any expenses when the mother or the child need to be hospitalised (e.g. in case of malnutrition) | NGOs |

**Table 4. Summary of recommendations to improve child and maternal nutrition.**

| Recommendations | Expected outcome | Themes |
|---|---|---|
| 1. Improve food availability and accessibility | Expected positive effect on dietary diversity and quantity of food consumed by mothers and children. | Livelihood resources & Natural resources |
| 2. Targeted provision of 'good foods' to lactating and pregnant mothers, including millet porridge, greens, beans, dodo, cassava leaves, meat and fish | Expected increased milk production in lactating mothers with benefits for child nutrition. | Home/Family environment & Livelihood strategies |
| 3. Assess and redress barriers related to insecure land tenure and poor land quality in communities | Expected increased food production. | Livelihood resources |
| 4. Promote increased economic and social support from men/partners during breastfeeding and complementary feeding period | Expected decreased level of stress in mothers, increased household food availability, and increased milk production and the possibility to breastfeed longer. | Health services, Family environment, Community environment & Economy |
| 5. Support and treatment to address the drivers and consequences of alcoholism in Batwa and Bakiga families | Expected decreased level of stress in mothers, increased household food availability, and increased milk production and the possibility to breastfeed longer. | Family environment, Community environment & Public policy environment |
| 6. Targeted training and rural outreach to support complementary feeding practices, potentially in conjunction with immunisation/antenatal medical services and trained traditional birth attendants | Expected improvement in complementary feeding practices with better child nutrition outcome. | Health services, Family environment, Community environment |
| 7. Strengthening family planning services | Expected more awareness when making the decision of conceiving, taking into account the importance of breastfeeding longer before giving birth to another baby. | Health services |
| 8. Promote sustainable agriculture, including taking into account smallholder practices and local knowledge. | Expected better adaptation to extreme climatic events to improve food production and availability, which could benefit mothers and children. | Livelihood strategies |

foods and malnutrition, importance of giving birth at the hospital and vaccinating children, family planning, and HIV testing before delivery. The participants reported that some information on child nutrition was also available through radio programmes that could be easily accessed by Batwa and Bakiga families. Some mothers reported receiving information from peers, local religious leaders, and at school (Batwa and Bakiga FGDs).

Knowledge on breastfeeding and HIV was limited, and participants mentioned that breastfeeding should be stopped at six months if the mother is HIV positive. Also, they reported that if the child has teeth, it is better to start food sooner to avoid any bites on the breast that could facilitate the spread of HIV from mother to child. They also added that this is a suggestion given by health professionals (Batwa and Bakiga FGD).

**Cultural beliefs.** Mothers discussed how cultural beliefs and local knowledge influence choices during breastfeeding and the complementary feeding period. Specifically, they discussed the role of traditional birth attendants (TBAs), and the practice of 'chewing' food for the child to make the first solid food softer.

Mothers reported confidence in the expertise of TBAs, including their knowledge on breastfeeding and complementary feeding, and willingness to be supported by them for antenatal and postnatal care together with the health care workers:

*"We like TBAs because they are cheaper than the hospital, they help us during delivery and they give us advice on breastfeeding and solid foods. Also, they do not shout at us if we don't bring clothes for the baby as happens in the hospital [. . .]. Also, there is no transport cost, and they cure (malnutrition) much faster with local plants. Indeed, when you go to the hospital you stay at least 1 week [. . .]."* (Batwa FGD).

Some Batwa participants mentioned also that TBAs offer traditional herbs to help with milk production, especially when mothers cannot buy millet porridge or do not have enough food

to eat; therefore, mothers approach TBAs to ask for help with breastfeeding. A Mutwa mother also added that according to traditional practice, mothers are expected to stay at home in the first four days after birth to produce more milk: "*If you do not stay at home, you won't produce milk*".

Mothers discussed other practices used in the past to help children eat first solid foods, such as 'chewing': "*This is not in practice anymore to avoid the diseases that came here [. . .]. Those changes (type of food and chewing practice) had an impact on child growth, children now grow better*" (Batwa FGD). However, many Batwa and Bakiga women did not support this position, arguing that children were healthier in the past when mothers used to chew wild foods from the forest before feeding the chewed food to the infant.

**Environment changes and environmental change.**   Weather variability, shifting seasonality, and environmental factors were discussed by the mothers as playing a role in the success or failure of breastfeeding and complementary feeding practices. Participants perceived that due to climate change, the seasons '*are not the same as before*' and production of food has decreased, with consequences on maternal and child health status and nutrition:

> "*[It is a] hard to time to grow crops as unexpected heavy rainfalls or droughts can come, and cause famine. This has an impact on children's growth*" (Bakiga FGD).

> "*There has been a change in the weather: food is available in harvesting seasons, but less than before. Sometimes it rains a lot, and crops are destroyed [. . .]. We don't have enough food and our bodies become weak*" (Batwa FGD).

> "*Land is overcultivated and food is scarce now, without a good rainy season there is not enough food for children*" (Bakiga FGD).

Also, participants perceived that a rise in temperatures has spread new diseases, such as malaria that before was uncommon, and due to this, children eat less, and get malnourished easily:

> "*It is hotter than before and new diseases [such as] malaria are affecting mothers and children. [. . .] Children get sick, and often undernourished*" (Bakiga FGD)

Further, women stated that 'December, January' and 'May and June' are the harvesting months, the preferred time to introduce solid food, but '*these seasons may not be always good*' anymore; therefore, the introduction of complementary feeding may be started later than 6 months, and weaning may be postponed:

> "*We can wait until [the child is] 1 year and a half or 2 years [old], but we need to wait for good times, target the best season when crops are grown, otherwise children become sick and malnourished very easily*" (Bakiga FGD).

Mothers also reported that the type of food consumed during the complementary feeding period is 'not energetic' as wild meat is not consumed or consumed rarely. Participants argued that some families depend on only one type of food for the whole week because of food unavailability (Bakiga FGD), and they mostly eat once or twice a day only. A Mukiga mother discussed the consequences of extreme weather events on child nutrition and growth, adding that environmental changes contributed to food insecurity and reduced breastmilk production:

*"If you do not have food due to too much sunshine that causes droughts, you cannot feed the child well. If there are droughts and dust, this can lead to sicknesses, and limits the mother to breastfeed the child well. When there is not food, you cannot express enough breastmilk, so the child is not satisfied"* (Batwa FGD).

## Discussion

This study explored the key factors affecting breastfeeding and complementary feeding among the Batwa and Bakiga communities in south-western Uganda. Other studies in Uganda have only quantitatively investigated factors affecting complementary feeding by using Uganda Demographic Health Survey (UDHS) data focussing on child health, vaccination status and maternal education [49,50], and data on Indigenous communities are scarce. Our mixed-methods design allowed to explore the socio-cultural and environmental barriers to child feeding practices among the Indigenous Batwa population for which there are no data on the UDHS. Also, we compared the information with the neighbouring Bakiga who shared a similar experience in terms of breastfeeding and complementary feeding pratices.

Additionally, in collaboration with the community we synthetisized recommendations to improve breastfeeding and complementary feeding practices among Batwa and Bakiga communities (Table 4). These included suggestions to enhance the food environment, such as food availability and accessibility, health services, and community and medical support as summarised in Table 4. The community, also, recognized barriers and opportunities across levels and themes identified in our framework (Fig 1).

The Batwa and Bakiga are aware of the consequences that unavailability of food can have on maternal and child health. A study conducted in 2019 showed that the diet of Batwa and Bakiga communities is low in proteins and fats, and the caloric content of the most commonly consumed foods is low or very low, with negative implications for child nutrition [28]. For these reasons, women highlighted the need for the District and Government to improve access to, and security and ownership of, more land to cultivate, and distribution of food to pregnant and lactating mothers and children.

The issue of lack of land, land ownership and access, and loss of Indigenous food is linked to the eviction in 1991 from the ancestral forest; this phenomenon is also common in many other Indigenous communities globally [24,51,52]. Assessing barriers to land access, insecure land tenure and being supported by the Government are priorities to increase food production and improve children' nutritional status.

Lack of representation and socio-economic marginalisation of Indigenous communities persists in the region; the Batwa lack representation at political levels, have limited voice in political decisions, perceive disenfranchisement from local leaders, and need to compete with non-Batwa to be elected to leadership positions [29]. Batwa and Bakiga women are especially marginalised and are victims of domestic violence. Marginalisation is linked to poverty [53–55]; the Indigenous Batwa communities are among the poorest communities in the world [18]. Studies have shown how wealth index and education level influences nutritional status [56,57]; indeed, both Batwa and Bakiga populations have extremely low education levels and income, and suffer from undernutrition and stunting, especially in children under 5 years [20].

Similarly, studies have demonstrated that increasing female empowerement can be beneficial to improve nutritional and caloric intake, especially dietary diversity, in infants and children [58,59]. Batwa and Bakiga participants reported that domestic violence is common among the communities, suggesting that empowerement levels remain low, and support from —and for—the community is required. Alcoholism, linked to domestic violence, is a factor affecting breastfeeding at individual, group and societal level within our framework, and has

been reported in other low income settings [53,60]. Alcohol is closely linked to poverty and food insecurity, with alcohol use perceived as partially driven by the desire to quell hunger pains. Women noted that being stressed by a non-supportive and alcohol-consuming partner coupled with food insecurity were some of the main factors associated with inadequate production of milk. For this reason, social and economic support from partners, and engagement from the district and Government to address the consequences of alcoholism in the community were identified as key needs. Indeed, research has shown that support from partners and family had, in fact, a positive effect on breastfeeding as mothers reported to be more confident and motivated [61]. Also, laws and regulations to support victims of abuse are needed at local level; the aim is to ensure an environment where women can safely nourish their children.

Additionally, women reported the importance of professional support at the hospital similarly to other research conducted in Kenya [62]. From the health care providers, mothers expected to get more information on feeding practices, especially complementary foods, and support to access family planning services to breastfeed longer. Previous research has shown that getting information and support at the health facility is not easy during antenatal visits due to the number of patients and amount of work of nurses and midwives [62]. Benefits from creation of breastfeeding and complementary feeding support groups led by community workers, volunteers or outreach nurses and midwives have been demonstrated in literature, and may be appropriate in this region and context [63].

However, participants also highlighted the important role of TBAs to provide essential newborn care in the communities, including advice on breastfeeding and complementary foods. The role of TBAs has long been discussed, although their potential contribution in helping mothers during the delivery and afterwards has been recognized in research [64]. In Uganda, there was a transition from the promotion of skilled birth attendants to the ban of their involvement during deliveries in according to the recommendation of WHO and the Safe Motherhood initiative [65]. However, TBAs continue to look after around 50% of pregnant mothers, especially in remote areas [66]. Studies in some low-income countries have suggested that trained TBAs can help remote communities in accessing health services, and giving support to families by promoting effective neonatal care [67,68]. Therefore, there is the need for health workers to collaborate with TBAs in order to encourage culturally accepted care for the rural and Indigenous communities in the hospitals and health centres in south-western Uganda.

Our study also found that environmental variability and change can play a role in food insecurity by affecting breastfeeding and complementary feeding practices. For example, according to the participants, seasonality was taken into account when introducing the first solid foods; in fact, complementary feeding was found to be delayed or anticipated depending on the season, and availability of food in the household. The time of introduction of solid foods impacts child nutrition and nutritional status, especially if the food insecurity persists for long periods [69]. Studies conducted in the same area show linkage between environmental factors and maternal and neonatal health. For example, *Bryson et al.* [25] described the link between pregnancy outcomes and climate change in Kanungu District among Indigenous and non-Indigenous populations, highlighting the impact of environmental factors on food security and maternal diet. Also, *MacVicar et al.* [22] found a causal pathway between weather, seasonal variability and birth weight among the same communities. These findings suggest that changing climate may have an impact on breastfeeding and complementary feeding practices through pathways of environmental variability and food security in the region.

Research has demonstrated that promoting sustainable agriculture is key to maintain and/ or increase food security in a changing climate [70]. Sustainable agriculture has been identified as an important climate response with the potential for double benefits: potential reduction in

emissions and adaptation to the implications of climate change for food security [71]. Common principles of sustainable farming include intercropping, high plant diversity and seed recycling [72]. Efficient adaptation strategies to climate change stressors on agricultural productivity will be important in underpinning nutritional outcomes for Batwa and Bakiga mothers and children during the breastfeeding and complementary feeding period. However, transitioning to climate-smart or sustainable agriculture in many cases requires a strong knowledge base, financial resources, and adequate policies; for populations in extreme poverty such as the Batwa–where baseline development remains severely inadequate – transitions to sustainable pathways are unlikely to be feasible without substantial and targeted support [73].

## Strengths and limitations

This study followed *Hector et al.* 's breastfeeding model [13], which is used to plan health interventions, and was adapted to investigate not only breastfeeding, as in the original framework, but also complementary feeding practices. The use of a mixed methods design and a community-based approach enriched the quantitative data with qualitative narratives, describing in detail key factors affecting child nutrition; also, group discussions offered the opportunity to share experiences and compare situations of lactating mothers living in the community. In spite of a relatively low number of FGDs (n = 12) conducted among the communities, we reached data saturation.

Although in the FGDs may be difficult for all participants to raise their voice due to a few dominant personalities in the group and power imbalances [74], the local research team had worked for many years with the communities and knew how to engage with the participants, mitigating this issue by making sure that everyone had the possibility to talk. Despite the expertise of the local team, there is a possibility that the researchers' positionality may have influenced the results [75]. To counteract this, the researchers ensured that interviews were conducted in a safe and confidential environment. Participants discussed sensitive topics such as domestic violence and alcoholism or mental health related-issues without any specific question on this matter because they were eager to highlight the problem, share their experience, and think of possible solutions.

Another strength of this study is the addition of the environmental level to the framework which allowed us to explore the relationship between weather and extreme climatic events with child feeding practices. Interesting recommendations have been suggested by the participants, and they may be useful for interventions in rural Uganda and other low-income regions where food insecurity and malnutrition are exacerbated by climatic changes [76–79].

The findings of this research, coming from a minority group, contribute to the research field of global public health and nutrition among Indigenous communities with a focus on women and children, and they will aid in tailoring nutrition interventions to specific local community's needs. Future research exploring breastfeeding and complementary feeding practices in the area may involve fathers, TBAs and health workers to investigate different perspectives, and to create more awareness on child nutrition and feeding practices among men. Indeed, the focus on IYCF, and in particular on the first 1000 days of life, is needed to prevent malnutrition and ensure that the children grow healthy and fully develop their capacities into adulthood [7].

## Supporting information

**S1 Data. Coordinates of settlements in Kanungu District.**
(XLSX)

**S1 Text. Individual interviews guide questions.**
(DOCX)

**S1A Text. Individual interviews guide questions in Rukiga language.**
(DOCX)

**S2 Text. Focus group discussions guide questions.**
(DOCX)

**S2A Text. Focus group discussions guide questions in Rukiga language.**
(DOCX)

**S3 Text. Inclusivity in globa research questionnaire.**
(DOCX)

## Acknowledgments

This research was possible thanks to the Batwa and Bakiga communities' contribution, who actively participated to the interviews and FGDs, and gave recommendations to improve child feeding practices. Special thanks to Grace Asaasira, an important member of our research team and the Batwa community, who is highly missed.

## Author Contributions

**Conceptualization:** Giulia Scarpa.

**Data curation:** Giulia Scarpa.

**Formal analysis:** Giulia Scarpa.

**Funding acquisition:** Giulia Scarpa.

**Investigation:** Giulia Scarpa.

**Methodology:** Giulia Scarpa.

**Project administration:** Sabastian Twesigomwe.

**Resources:** Sabastian Twesigomwe, Paul Kakwangire, Ester Nowembabazi, Charity Kesande.

**Supervision:** Lea Berrang-Ford, Janet E. Cade.

**Validation:** Giulia Scarpa.

**Writing – original draft:** Giulia Scarpa.

**Writing – review & editing:** Giulia Scarpa, Lea Berrang-Ford, Sabastian Twesigomwe, Paul Kakwangire, Maria Galazoula, Carol Zavaleta-Cortijo, Kaitlin Patterson, Didacus B. Namanya, Shuaib Lwasa, Ester Nowembabazi, Charity Kesande, Janet E. Cade.

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
