## [Decision Letter · Decision Letter 0]

26 Oct 2021

PGPH-D-21-00745

Socio-economic and environmental factors affecting breastfeeding and complementary feeding practices among Batwa and Bakiga communities in south-western Uganda.

Dear Dr. Giulia Scarpa,

Thank you for submitting your manuscript to PLOS Global Public Health. After careful consideration, we feel that it has merit but does not fully meet PLOS Global Public Health’s publication criteria as it currently stands. Therefore, we invite you to submit a revised version of the manuscript that addresses the points raised during the review process.

We look forward to receiving your revised manuscript.

Kind regards,

Srinivasa Rao Mutheneni, PhD

Academic Editor

Journal Requirements:

1. Please include a complete copy of PLOS’ questionnaire on inclusivity in global research in your revised manuscript. Our policy for research in this area aims to improve transparency in the reporting of research performed outside of researchers’ own country or community. The policy applies to researchers who have travelled to a different country to conduct research, research with Indigenous populations or their lands, and research on cultural artefacts. The questionnaire can also be requested at the journal’s discretion for any other submissions, even if these conditions are not met.  Please find more information on the policy and a link to download a blank copy of the questionnaire here: https://journals.plos.org/plosone/s/best-practices-in-research-reporting. Please upload a completed version of your questionnaire as Supporting Information when you resubmit your manuscript.

2. Please include additional information regarding the survey or questionnaire used in the study and ensure that you have provided sufficient details that others could replicate the analyses. For instance, if you developed a questionnaire as part of this study and it is not under a copyright more restrictive than CC-BY, please include a copy, in both the original language as well as the English version already provided, as Supporting Information.

3. You indicated that you had ethical approval for your study. In your Methods section, please ensure you have also stated whether you obtained consent from parents or guardians of the minors included in the study or whether the research ethics committee or IRB specifically waived the need for their consent.

4. Please provide separate figure files in .tif or .eps format only and remove any figures embedded in your manuscript file. Please ensure that all files are under our size limit of 20MB.

Once you've converted your files to .tif or .eps, please also make sure that your figures meet our format requirements.

5. Please provide us with a direct link to the base layer of the map used in Fig 2 and ensure this location is also included in the figure legend. 

Please note that, because all PLOS articles are published under a CC BY license (creativecommons.org/licenses/by/4.0/), we cannot publish proprietary maps such as Google Maps, Mapquest or other copyrighted maps. If your map was obtained from a copyrighted source please amend the figure so that the base map used is from an openly available source.

Please note that only the following CC BY licences are compatible with PLOS licence: CC BY 4.0, CC BY 2.0  and CC BY 3.0, meanwhile such licences as CC BY-ND 3.0 and others are not compatible due to additional restrictions. If you are unsure whether you can use a map or not, please do reach out and we will be able to help you. 

The following websites are good examples of where you can source open access or public domain maps:

6. We have noticed that you have uploaded supporting information but you have not included a list of legends.  Please add a full list of legends for all supporting information files (including figures, table and data files) after the references list.

Additional Editor Comments (if provided):

Reviewers' comments:

Reviewer's Responses to Questions

**Comments to the Author**

1. Does this manuscript meet PLOS Global Public Health’s publication criteria? Is the manuscript technically sound, and do the data support the conclusions? The manuscript must describe methodologically and ethically rigorous research with conclusions that are appropriately drawn based on the data presented.

Reviewer #1: Yes

Reviewer #2: Yes

2. Has the statistical analysis been performed appropriately and rigorously?

Reviewer #1: Yes

Reviewer #2: Yes

3. Have the authors made all data underlying the findings in their manuscript fully available (please refer to the Data Availability Statement at the start of the manuscript PDF file)?

Reviewer #1: Yes

Reviewer #2: Yes

4. Is the manuscript presented in an intelligible fashion and written in standard English?

Reviewer #1: Yes

Reviewer #2: Yes

5. Review Comments to the Author

Reviewer #1: The authors have written the paper very well and it is technically sound with relevant literature to support the study. Also, they have been able to use the results with back up of literature in the discussion section to support their conclusion. The authors have also been able to point out some key areas to be considered in future research in order to improve breastfeeding and complementary feeding practices in the study area (south-western Uganda).

Statistical analysis has been performed appropriately and rigorously. Also, the manuscript is presented in an intelligible fashion and written in standard English. However, there are some areas that require minor corrections as follows:

a) Line 68: ………. “and food behaviour among Indigenous families due to changing in socio-environmental”; replace the word “changing” with “changes”.

b) Line 174: “Fig.2: In the map we represented the ten Batwa settlements that participating in the research; there is….” replace the word “participating” with “participated”.

c) Line 183: “Kihembe/Kengoma cell), 2) two settlements closed to the forest and located…..” replace the word “closed” with “close”.

d) Line 202: “singular) female researcher. The FGDs and the individual interviews were conducted and audio-recorded in….” – write the FGDs in long form since it is the first time it is used in the main text and delete “The”. Therefore, line 202 will read: “singular) female researcher. Focus group discussions (FGDs) and the individual interviews were conducted and audio-recorded in….”.

e) Line 206: “We created an interview guide with open questions for the focus group discussion (Appendix 1).” – replace focus group discussion with FGD.

f) Line 207: “followed the themes from the Optimal Infant and Young Child Feeding guidelines,” – replace Infant and Young Child Feeding with IYCF, because it has been written in long form in line 53.

g) Lines 254-256: “Mothers of all children older than 7 days reported that their child had experienced one or more episodes of illness in the first 6 months of life that had had a negative impact on breastfeeding.” It is not clear if the authors mean 7 days OR 7 months. I suggest to authors to re-check on this.

h) Lines 393-396: “Information on child feeding is provided to mothers by the hospital or NGOs (Tab. 4; the main topics covered in these sessions includes breastfeeding practices, complementary foods and malnutrition, importance of giving birth at the hospital and vaccinating children, family planning, and HIV testing before delivery.” I suggest to authors to close a bracket after 4, i.e., (Table 4); the main topics………

i) Line 407: I suggest to bold the subheading “Cultural beliefs” for consistency.

j) Lines 475-477: “Participants did not limit their recommendations to improved food security only, however, and recognized barriers at opportunities across levels identified in our framework (Figure 1).” This sentence is not clear in its last part underlined. I suggest to the authors to consider rephrasing it to read as follows: “Participants did not limit their recommendations to improved food security only, but they recognized barriers and opportunities across levels identified in our framework (Figure 1).”

k) Line 503: “consequences of alcoholism in the community were identified as key needs). Indeed, research has….” – the bracket after “needs” is not needed; just delete it.

l) Line 517: “However, participants also highlighted the important role of traditional birth attendants (TBAs) to provide….” - delete “traditional birth attendants” and use only TBAs without brackets because it has already been written in long form and abbreviation in Line 409.

m) Line 535: “and maternal and neonatal health. For example, Bryson et al. (25) described the linkbetween pregnancy” – separate the word “linkbetween” to be two words.

n) Line 566: “and complementary feeding practices in the area may involve fathers, traditional birth attendants and health” – replace “traditional birth attendants” with TBAs

o) Line 568: “practices among men. Indeed, the focus on infant and young child feeding, and in particular on the first 1000” – replace “infant and young child feeding” with IYCF.

p) Line 573: “participated to the interviews and focus group discussions, and gave recommendations to improve child feeding” – replace “focus group discussions” with FGDs.

q) Figure 2 (page 36): “Fig.2: In the map we represented the ten Batwa settlements that participating in the research; there is a correspondent Bakiga settlement for each Batwa settlement. In our study, we involved……….” – replace “participating” with “participated”.

Reviewer #2: I commend the authors for a well done write up that is easy to understand while tackling an important public health problem. I have a few suggestions:

Major comments:

1) there seems to be a disconnect between the data presented on Table 1 on birth interval and the claims from the FDG that Batwa women get pregnant fast on line 370 kindly qualify that and probably link that to teen pregnancies

2)The data provided seemed to have only one set of twins yet your findings though resonable provide strong claims on transistions for weaning practices. I would suggest that you update this to also include competition of food resources for young children with many siblings <5 years to strengthen the claim

3) I suggest adding a short paragraph about recommendations after the results and move table 3 from the discussion section below the new paragraph. Also improving food availability and accessibility is a very generic recommendation in this case, instead when writing the paragraph mention that recommendations targeting improvement of foood availability and accessibility were identified as summarised on Table 3.

4) On the table 3, levels are confusing, instead replace with themes that emerge from your study such as support etc.

5) I would suggest adding an additional recommendation on climatic changes if in this communities sustainable farming such as irrigation and production of drought resistance crops as well as improving drainage through contours are feasible to counter effects of floods and droughts. Kindly review available literature to support this arguement.

6)On discussion kindly expand paragraph 1 to showcase what are the deviations and similarities of factors identified from studies identified by UDHS to support your claim on focussing on indigenous communities whose issues may be overlooked from UDHS findings. Also define UDHS in full not every reader can relate that you are talking of Uganda demographic Health Surveys.

Minor comments:

Remove the spacing after pages 6 and 23

On line 535 space the words "likbetween"

On line 48 put a comma after 2019

Lable tables in a standard manner throughout text instead of Tab write in full Table

6. PLOS authors have the option to publish the peer review history of their article (what does this mean?). If published, this will include your full peer review and any attached files.

**Do you want your identity to be public for this peer review?** For information about this choice, including consent withdrawal, please see our Privacy Policy.

Reviewer #1: **Yes: **Eliudi Saria Eliakimu

Reviewer #2: No

---

## [Decision Letter · Decision Letter 1]

19 Jan 2022

Socio-economic and environmental factors affecting breastfeeding and complementary feeding practices among Batwa and Bakiga communities in south-western Uganda.

PGPH-D-21-00745R1

Dear Dr. Giulia Scarpa

We're pleased to inform you that your manuscript has been judged scientifically suitable for publication and will be formally accepted for publication once it meets all outstanding technical requirements.

Within one week, you'll receive an e-mail detailing the required amendments. When these have been addressed, you'll receive a formal acceptance letter and your manuscript will be scheduled for publication.

An invoice for payment will follow shortly after the formal acceptance. To ensure an efficient process, please log into Editorial Manager at https://www.editorialmanager.com/pgph/ click the 'Update My Information' link at the top of the page, and double check that your user information is up-to-date. If you have any billing related questions, please contact our Author Billing department directly at authorbilling@plos.org.

Kind regards,

Srinivasa Rao Mutheneni, PhD

Academic Editor

Additional Editor Comments (optional): Nil

Reviewers' comments: Nil

Reviewer's Responses to Questions

**Comments to the Author**

1. If the authors have adequately addressed your comments raised in a previous round of review and you feel that this manuscript is now acceptable for publication, you may indicate that here to bypass the “Comments to the Author” section, enter your conflict of interest statement in the “Confidential to Editor” section, and submit your "Accept" recommendation.

Reviewer #1: (No Response)

Reviewer #2: All comments have been addressed

2. Does this manuscript meet PLOS Global Public Health’s publication criteria? Is the manuscript technically sound, and do the data support the conclusions? The manuscript must describe methodologically and ethically rigorous research with conclusions that are appropriately drawn based on the data presented.

Reviewer #1: Yes

Reviewer #2: Yes

3. Has the statistical analysis been performed appropriately and rigorously?

Reviewer #1: Yes

Reviewer #2: Yes

4. Have the authors made all data underlying the findings in their manuscript fully available (please refer to the Data Availability Statement at the start of the manuscript PDF file)?

Reviewer #1: Yes

Reviewer #2: Yes

5. Is the manuscript presented in an intelligible fashion and written in standard English?

Reviewer #1: Yes

Reviewer #2: Yes

6. Review Comments to the Author

Reviewer #1: The authors have addressed well all the comments I provided EXCEPT one comment quoted below: ""b) Lines 254-256: “Mothers of all children older than 7 days reported that their child had experienced one or more episodes of illness in the first 6 months of life that had had a negative impact on breastfeeding.” It is not clear if the authors mean 7 days OR 7 months. I suggest to authors to re-check on this. THIS COMMENT HAS NOT BEEN ADDRESSED (((in the Revised Manuscript [PGPH-D-21-00745R1] it is in lines 252-254))). The fact that the mothers reported that their children had experienced one or more episodes of illness in the first 6 months, suggest that it should read """Mothers of all children older that 7 MONTHS......"""

Reviewer #2: All the comments and suggestions have been updated accordingly and the manuscript reads well.

7. PLOS authors have the option to publish the peer review history of their article (what does this mean?). If published, this will include your full peer review and any attached files.

**Do you want your identity to be public for this peer review?** For information about this choice, including consent withdrawal, please see our Privacy Policy.

Reviewer #1: **Yes: **Eliudi Saria Eliakimu

Reviewer #2: No
